# Self-Administration of Meloxicam via Medicated Molasses Lick Blocks May Improve Welfare of Castrated Calves

**DOI:** 10.3390/ani15030442

**Published:** 2025-02-05

**Authors:** Samantha Rudd, Sabrina Lomax, Peter J. White, Dominique Van der Saag

**Affiliations:** 1Sydney School of Veterinary Science, Faculty of Science, The University of Sydney, Sydney, NSW 2006, Australia; p.white@sydney.edu.au (P.J.W.); dominique.van.der.saag@sydney.edu.au (D.V.d.S.); 2School of Life and Environmental Sciences, Faculty of Science, The University of Sydney, Sydney, NSW 2006, Australia; sabrina.lomax@sydney.edu.au

**Keywords:** castration, cattle, inflammation, lick block, meloxicam, non-steroidal anti-inflammatory drug, pain, self-administration

## Abstract

Livestock husbandry procedures inflict long-term pain and inflammation, with implications for animal welfare and production. Current formulations of pain relief available to livestock producers, including topical anaesthetics and non-steroidal anti-inflammatory drugs, are limited in their duration of action, only lasting 24–72 h. The ad libitum administration of the non-steroidal anti-inflammatory drug, meloxicam, orally via a medicated lick block is a novel concept that may enable pre-emptive and long-term analgesia to be established and maintained in a manner that is practical for livestock producers. This novel delivery system was evaluated against conventional subcutaneous meloxicam in surgically castrated calves. Results showed that therapeutic concentrations of meloxicam were established in calves fed medicated lick blocks; however, concentrations were highly variable between individuals. The administration of meloxicam by this method improved wound healing scores. Meloxicam, regardless of administration method, improved some behavioural outcomes following surgical castration. The results of this study highlight the complexities of assessing and addressing pain in cattle. If shown to be safe, medicated lick blocks could be a feasible option for providing improved pain relief to cattle, leading to advancements in animal welfare.

## 1. Introduction

Castration of male animals is a practice commonly performed in livestock production systems and involves the removal or destruction of the testes. The procedure enables the diversion of energy from the maturation of sexual organs and production of germline cells to physical growth and performance [1], improves meat quality by increasing tenderness and reducing dark cutting [2], and reduces aggression, which improves safety for both cattle and handlers [3,4]. Although there are long-term and justifiable benefits to castration, the procedure is highly invasive, causing pain, inflammation, and infection, regardless of method used [5,6,7]. Castration is therefore an animal welfare concern that needs refining in the interim until there are viable alternatives to conducting the procedure.

The provision of appropriate pain relief is now recognised as essential in meeting welfare requirements of animals. The Australian Animal Welfare Standards and Guidelines for Cattle state that pain relief should be provided to livestock undergoing invasive husbandry procedures when aged over 6 months or over 12 months if at first yarding in some jurisdictions [8]. The Animal Welfare Standards and Guidelines for Cattle also acknowledge that pain relief drugs that are currently available for use in cattle are limited in their effectiveness and duration [8]. Multi-modal pain relief, usually local anaesthetic in combination with a non-steroidal anti-inflammatory drug (NSAID), has been determined as the most effective method of reducing both acute and inflammatory pain associated with husbandry procedures [7,9,10].

The topical anaesthetic Tri-Solfen^®^ (Dechra Veterinary Products (Australia) Pty. Ltd., Somersby, NSW, Australia) is a commonly used form of pain relief for husbandry procedures in livestock. Tri-Solfen is composed of two anaesthetic products (lignocaine and bupivacaine), a vasoconstricting agent (adrenaline), and an antiseptic (cetrimide) [11]. Tri-Solfen^®^ is applied post-procedure, sprayed directly onto the wound, and is absorbed through mucous membranes, inducing anaesthesia and vasoconstriction [12]. For surgical castration wounds, Tri-Solfen^®^ is sprayed directly into the scrotum and onto the retracting spermatic cord post-operatively and has been shown to reduce post-operative pain in sheep and cattle for up to 24 h [12,13]. Due to its post-operative application, Tri-Solfen^®^ does not address the procedural pain, nor the long-term pain associated with husbandry procedures such as surgical castration. Therefore, the use of a multi-modal approach is critical for effective pain relief [7,9,10].

Meloxicam is a commonly used NSAID in cattle, available by veterinary prescription in Australia as a subcutaneous (SC) or intramuscular (IM) injection or via oral trans mucosal (OTM) (Ilium^®^ Buccalgesic, Troy Laboratories, Glendenning, NSW, Australia) or oral suspension (Meloxi-care, Parnell, Alexandria, NSW, Australia) delivery. Meloxicam has been shown to successfully reduce inflammation and pain associated behaviours in castrated cattle for up to 72 h [6,14]. Meloxicam is preferred over other NSAIDs for its long half-life of 27 h in cattle (oral administration) and selectivity to cyclooxygenase (COX)-2 [14,15]. Selectivity to COX-2 is important for targeted pain relief, as COX-2 is mostly expressed in damaged tissue [16]. While the duration of action of meloxicam is one of the longest among available NSAIDs, the duration of pain and inflammation resulting from surgical husbandry procedures such as castration likely persists for up to several weeks [17]. Current methods of drug administration are invasive and require restraint of the animal. Administration of multiple drug doses is impractical, stressful to animals, and has potential to cause further damage to healing wounds during yarding and handling.

The self-administration method of delivering NSAIDs to animals via feed or oral supplements is a relatively new concept that allows the drug to be administered pre- and post-operatively, independent of human contact and handling. The delivery of the NSAID flunixin meglumine via medicated pellets was shown to reach maximum therapeutic plasma drug concentration by an average of 6 h after consumption [18] and a reduction in pain and inflammation associated with castration and tail-docking of lambs [19]. Comparative pharmacokinetic studies of oral versus IM delivery of meloxicam show that the bioavailability is often reduced when orally administrated, requiring increased dosage from 0.5 mg/kg SC to 1.0 mg/kg [18,20,21,22]. The elimination half-life of oral meloxicam (28.5 h) is significantly longer than SC meloxicam (12.5 h), at the recommended dosage rates of 1.0 mg/kg and 0.5 mg/kg, respectively, extending the duration of action [21,23].

The use of lick blocks in cattle production systems enables the provision of various supplements such as vitamins and minerals via a non-invasive and external source [24]. The aim of this study was to determine if the delivery of meloxicam via a medicated molasses lick block would result in therapeutic plasma drug concentrations and provide analgesia following surgical castration in beef calves. It was hypothesised that calves that were provided ad libitum access to a molasses lick block medicated with meloxicam would have therapeutic plasma concentrations of the drug maintained over the period of exposure to the block, leading to reduced pain during this time.

## 2. Materials and Methods

An experiment was conducted using the University of Sydney’s commercial beef herd located in Greendale, NSW, Australia. Forty weaned *Bos taurus* mixed Angus and Charolais male calves, comprising 30 bulls and 10 steers (4 to 6 months of age), were enrolled in this study. The calves were bred on the property where the experiment was performed. Treatment groups were not blocked by breed. The calves had been weaned approximately 3 weeks prior to the experiment. During this 3-week period, calves were acclimated to the cattle yards, race, and crush and were provided ad libitum access to molasses lick blocks (Greenspell™, 4 Seasons Company Pty Ltd., Crestmead, QLD, Australia) to ensure they were consuming the lick. The experimental protocol was approved by the Animal Ethics Committee of the University of Sydney (Project number 2020/1804).

### 2.1. Treatments and Experimental Design

The experiment was conducted over 14 days with measurements taken on day −1 (1 day prior to treatment), 0 (day of treatment), and days 1, 2, 3, 6, 9, and 13 following treatment. The steers had been rubber-ring castrated within 48 h after birth and were completely healed before the experiment. An interactive program for performing power and sample size calculations (PS: Power and Sample Size Calculation) was used to determine the number of animals necessary for this project [25]. The primary outcome of the study, meloxicam concentration, is a continuous response variable from independent control and experimental subjects with 1 control per experimental subject. The true difference in the experimental and control means was predicted to be 20% and the standard deviation to be 15%. Therefore, 10 experimental subjects and 10 control subjects were required to be able to reject the null hypothesis, i.e., that the means of the experimental and control groups are equal with a probability (power) of 0.8. The Type I error probability associated with this test of this null hypothesis is 0.05. The steers were allocated to the first treatment group: (1) no castration (positive control; PC; *n* = 10). The bull calves were randomly allocated to 1 of 3 treatment groups as they presented in the race 10 at a time: (2) surgical castration with topical anaesthesia (negative control; NC; *n* = 10); (3) surgical castration with topical anaesthesia and subcutaneous meloxicam injection (Metacam20^®^ 20 mg/mL at 0.5 mg/kg; M; *n* = 10); and (4) surgical castration with topical anaesthesia and ad libitum access to two molasses lick blocks medicated with meloxicam (4 Seasons Company Pty Ltd., Crestmead, QLD, Australia; ML; *n* = 10). The concentration of meloxicam in the lick blocks was 1000 mg/kg, and the lick block weighed 39.85 kg in total. A previous unpublished pilot study conducted by the research team monitored lick block intake in a similar mob of calves to determine optimal lick block concentration. Prior to commencing the study, three random samples were taken from each lick block to ensure the meloxicam was still active. Calves in treatment groups 1, 2, and 3 were given ad libitum access to unmedicated molasses lick blocks (Greenspell™, 4 Seasons Company Pty Ltd., Crestmead, QLD, Australia) throughout the experiment. The calves were held on pasture in 0.2 ha pens in their respective treatment groups of 10 animals with ad libitum access to water and hay for the duration of the experiment. Calves had to be grouped by treatment to manage consumption of the lick blocks. The ML group were given access to the meloxicam lick blocks on day −1 and the medicated blocks were removed on day 6 and replaced with the unmedicated molasses lick blocks. On day −1 prior to castration, each calf was weighed using Gallagher^®^ cattle scales (Gallagher Group Ltd., Hamilton, New Zealand). At the same time, the calves were allocated an individual identification number from 1 to 40, spray-painted on their sides and back with cattle tail paint (Leader Products Pty Ltd., Craigieburn, VIC, Australia) for visual identification throughout the experiment. For treatment and data collection, calves were walked approximately 500 m from their pens to the handling facilities where they were restrained in a head bale within a cattle crush.

### 2.2. Castration and Pain Mitigation

All calves were administered an intramuscular sub-sedative dose of xylazine (Ilium Xylazil-20^®^, Troy Laboratories, Glendenning, NSW, Australia) at a rate of 0.05 mg/kg liveweight, a minimum of 5 min prior to castration by a veterinarian. This was injected into the rump of each calf while in the race before entering the crush to ensure time for standing sedation to take effect, identified by mild drooling, delayed ocular response, and slow movement. A sub-sedative dose was given so that the calves would remain standing during and following castration or sham castration. The calves were weighed the day previous to treatment (day −1). For castration, calves were restrained in the head bale, standing in the crush with the left hind leg tied forward using a rope to minimise movement. Approximately 1 mL of local anaesthetic lignocaine (Ilium Lignocaine-20^®^, Troy Laboratories, Australia) was infiltrated into the scrotal skin, and 2.5 mL injected into each spermatic cord and testis via the neck of the scrotum, a minimum of 3 min prior to castration. While restrained in the crush, calves in the M group were administered 0.5 mg/kg of meloxicam subcutaneously into the neck a minimum of 5 min before castration. The times of sedation and castration are detailed in Table 1.

All calves were surgically castrated by the same skilled technician using a sterile scalpel. The testes were held in the proximal section of the scrotum while a transverse incision was made around the entire circumference of the distal third of the scrotum, removing the tip. Each testis was individually pulled through the scrotum to expose the spermatic cord and associated blood vessels, which were then severed by scraping with the scalpel to reduce blood loss by promoting clotting. Five calves from the NC group were castrated with this method; however, due to the size of the testes, concern for haemorrhage prompted a change in method. The remainder of the calves were castrated by making an incision from the mid to distal end on the lateral sides of the scrotum of either testis. Each testis was held in the distal section of the scrotum by the technician to keep the skin taut and prevent the testis from retracting. The testes were removed in the same manner as previously described. Approximately 2 mL of topical anaesthetic and antiseptic gel (Tri-Solfen^®^, Dechra Veterinary Products (Australia) Pty. Ltd., Somersby, NSW, Australia) was administered directly into the scrotum onto the retracting spermatic cord tissue to induce anaesthesia and haemostasis [13].

### 2.3. Blood Sampling and Analysis

Blood samples were collected via jugular venepuncture from calves in the M and ML groups to determine plasma meloxicam concentration (PMC) and from all calves for serum haptoglobin concentration (PHC) analysis. The calves were restrained in the head bale of the crush for blood sampling, with their heads held to the side to access the jugular vein. Side of sampling was alternated daily with each sample. Samples were taken on day 0 from ML calves only, then all treatment groups on days 1, 2, 3, 6, 9, and 13. Blood samples were collected in 1 × 9 mL lithium heparin and 1 × 9 mL serum BD Vacutainer^®^ blood collection tubes (Becton, Dickinson and Company, Franklin Lakes, NJ, USA). Lithium heparin samples were immediately inverted several times to mix the anticoagulant with the blood and were stored on ice until centrifuging. Serum samples were not mixed to promote clotting and were left at room temperature for 30 min before being stored on ice until centrifuging. Samples were centrifuged at 1500 revolutions per minute for 10 min within 2 h after collection. The plasma/serum portion was then pipetted into 3 mL Eppendorf tubes as two aliquots per sample, labelled, then frozen at −20 °C immediately.

Plasma meloxicam concentration analysis was performed by high-performance liquid chromatography (HPLC) at the University of Sydney following the same methodology outlined in a previous study [26]. The lower limit of quantification was 0.048 µg/mL, and the upper limit was 25 µg/mL.

Serum haptoglobin analysis was performed using a Bovine Haptoglobin Immuno-peroxidase assay for Determination of Haptoglobin in Bovine Samples (MyBioSource, San Diego, CA, USA) at the University of Sydney. The lower limit of detection was 15 ng/mL, and the upper limit of detection was 1000 ng/mL.

### 2.4. Medicated Lick Block Weight Analysis

Medicated lick blocks provided to the ML group were weighed on days −1, 0, 1, 2, 3, and 6, then replaced with a non-medicated molasses lick block on day 6. The change in block weight was used to calculate the average amount of meloxicam consumed by the group of calves (Table 2).

The change in weight was calculated by subtracting each day’s weight from the previous day’s weight. Meloxicam consumption was calculated by multiplying the change in weight from each day by 1000, as the lick blocks contained 1000 mg/kg of meloxicam.

Average meloxicam consumption (mg/hd)$$=\frac{Total\;meloxicam\;consumption\;(mg)}{no.\;of\;animals\;(hd)}$$

Average meloxicam consumption per kilogram (mg/kg/hd)$$=\frac{average\;meloxicam\;consumption\;(mg/kg)}{average\;group\;weight\;\left(kg\right)}$$

Average meloxicam consumption (mg/kg/hd/day)$$=\frac{average\;meloxicam\;consumption\;(mg/kg)}{no.\;of\;days}$$

### 2.5. Scrotal Measurements

Scrotal diameter was measured using digital callipers (Craftright^®^, Bunnings Group Ltd., Burnley, VIC, Australia) at the widest part of the scrotum. Measurements were conducted on days 1, 2, 3 6, 9, and 13 on calves in the NC, M, and ML treatment groups.

### 2.6. Wound Images

Photographs of castration wounds were taken using a digital camera (EOS 700 D Canon Camera, Canon Australia Pty Ltd., Macquarie Park, NSW, Australia) with 24.1 MP resolution on days 1, 2, 3, 6, 9, and 13 for calves in NC, M, and ML treatments. These images were later used to score castration wounds based on degree of healing. Photographs were taken caudal to the scrotum as the calves were restrained in the crush with their left leg tied forward, providing a clear image of the entire scrotum. Wounds were scored on a numerical rating scale from 1 to 5 (Table 3). Determining wound scores via photographs was elected over live in-crush scoring to eliminate bias, as it was obvious which calves belonged to which treatment group while in the crush, but not in the photographs. This method of scoring wound healing from gross appearance has been used in multiple published studies [27,28,29,30,31]. Additionally, the use of photographs enabled wound scores to be determined by multiple observers. The observers were trained to use the wound-scoring scale during an online video call with the lead investigator and were blinded to treatment.

### 2.7. Wound Temperature

Infrared thermographs were taken using an infrared camera (FLIR^®^ E50 camera, Teledyne FLIR, Wilsonville, OR, USA) to measure wound temperature. Infrared thermographs were taken on days 1, 2, 3, 6, 9, and 13 of calves in treatment groups NC, M, and ML. As with the digital photographs, infrared thermographs were taken caudal to the scrotum as the calves were restrained in the crush with their left leg tied forward, providing a clear image of the entire scrotum. A square 20 × 20 cm frame was held over the wound site to ensure the distance of the camera from the wound was constant for accurate temperature validation. The frame was rested against the thigh of the calf and the camera frame was aligned with the top and bottom of the physical frame. The camera settings were calibrated every 30 min to the current ambient temperature and humidity. An infrared thermography software program (FLIR^®^ Tools Software (version 6.6.1807), Teledyne FLIR, Wilsonville, OR, USA) was used to determine the maximum temperature within each infrared thermograph, analysing the entire area within the frame.

### 2.8. Behavioural Observations

Calf behaviour was recorded via direct visual observation for the duration of the experiment. The same three trained observers performed all behavioural observations throughout the experiment and were blinded to treatment. Behaviours were recorded according to a customised ethogram adapted from other validated ethograms in the literature (Table 4) [13,29,31,32,33,34,35]. The observers walked around the perimeter of the pens to record 3 to 12 observations in total for each calf on each experimental day. Behaviours were recorded via instantaneous sampling at 30 s intervals. Behaviours were analysed as mutually exclusive and recorded binomially as either ‘0’ if the behaviour did not occur or ‘1’ if the behaviour did occur within the 30 s interval. Observations were conducted from 11:30 a.m. until 1:30 p.m. on days −1, 1, 2, 3, 6, 9, and 13 and from 11:30 a.m. to 1:30 p.m. (0 a.m.) and 2:10 p.m. to 4:15 p.m. (0 p.m.) on day 0.

### 2.9. Statistical Analysis

All data were collated in Microsoft Excel^®^ 2019 (Microsoft Corporation, Washington, DC, USA) and stacked for analysis. Statistical analyses were conducted using statistical software, Genstat^®^ 22nd edition (version 22.1.0.72, VSN International, Hemel Hempstead, UK) and R^®^ (version 4.4.2, R Foundation for Statistical Computing, Vienna, Austria).

Plasma meloxicam concentration (PMC), serum haptoglobin concentration (SHC), scrotal diameter and temperature, and percentage of change in weight (PCW) data were input into Genstat^®^ and checked for normality using the “distribution plots” function. This produced a Q-Q plot and Anderson–Darling (AD) statistics and critical values. Normality was determined if the AD statistic < AD critical value at 5% confidence. The PMC data were not normally distributed and were therefore transformed using Log_e_. A normality test of the transformed data was then performed using the “distribution plots” function, ensuring the transformed data were normally distributed. Outputs from the statistical analysis were then back-transformed using an exponential function in Microsoft Excel^®^ 2019 for presentation in Section 3. SHC data were also not normal but were unable to be successfully transformed due to the nature of the data, so the untransformed data were used for the analysis.

The PCW and scrotal diameter data were checked for outliers using the box-plot function in Genstat^®^. There were no outliers in the PCW data. There were multiple outliers in the scrotal diameter data; however, removal of relevant outliers only produced more outliers, so the data were analysed without removing outliers.

Plasma meloxicam concentration (transformed), SHC, scrotal diameter, scrotal temperature, and PCW were analysed using restricted maximum likelihood (REML) in Genstat^®^. Plasma meloxicam concentration in M and ML calves was also compared between individuals, scrotal temperature and diameter, and wound healing scores using REML. The fixed model for each test was treatment (PC, NC, M, and ML where applicable) x day (0, 1, 2, 3, 6, 9 and 13 where applicable). The random model used was calf identification number. A significant interaction between treatment and day was first tested, and non-significant terms were then dropped from the model to determine the effect of treatment and day. The model predicted means which were used for a pairwise comparison using least significant differences (LSDs). The predicted means and the standard errors were reported, along with their significant pairs. For all statistical tests, a *p* value of ≤0.05 was considered as significant, and 0.05 < *p* < 0.1 was considered a statistical trend.

Wound score data were stacked using Microsoft Excel^®^ 2019 and analysed using an ordinal logistic regression (OLR) in R^®^ within the ordinal package. An inter-observer reliability test was performed to test the wound-scoring system for repeatability between 3 observers, with 1 observer subsequently removed from further analysis. The wound scores of the NC calves were compared between the two castration methods, with the 5 calves castrated with the initial method subsequently removed from future wound analysis. The remaining 5 NC calves were compared with the M and ML calves. The fixed effects were treatment, day, and observer, and the random effect was the calf ID number. Post hoc analysis using Tukey’s comparisons was used to determine pairwise significance. For all analyses, a *p* value ≤ 0.05 was considered significant.

Behavioural data were recorded binomially (1 = occurred or 0 = did not occur) and stacked in Microsoft Excel^®^ 2019. Due to the low frequency of behaviours observed, abnormal and normal positional and locomotive behaviours were combined (i.e., normal lying and abnormal lying were amalgamated to “lying”). Similarly, head turning, tail flicking, leg movements including stomping or kicking, and abnormal positional and locomotive behaviours were amalgamated into a single behaviour and were referred to as “abnormal”. These behaviours had been previously validated for pain in the context of surgical castration by numerous authors [6,29,34]. Maintenance or comfort behaviours including eating, grooming, defecating, and ruminating were amalgamated with normal positional/locomotive behaviours and were analysed as “normal”. All behaviours were analysed by week due to the low frequency of daily behaviours, except for eating, which was analysed by observation timepoint. Observations prior to castration/sham castration (day −1) were referred to as “week −1”, observations made on days 0 to 3 were amalgamated to “week 1”, and observations made on days 6–13 were amalgamated to “week 2”. Behavioural data were analysed using a generalised linear mixed model (GLMM) in GenStat^®^ with a binomial distribution to determine the effect of treatment and timepoint (day or week). The model to be fitted was treatment (PC, NC, M, and ML) x day (−1, 0, 1, 2, 3, 6, 9, and 13) or week (−1, 1, 2). Behaviours were first tested for a significant interaction between treatment and day, and non-significant terms were then dropped from the model. The predicted proportions represent the percentage of time spent performing the behaviour. Least significant differences were used for pairwise comparisons of model predicted means to determine significance using Microsoft Excel^®^ 2019 and were presented with the standard error of the means (s.e.m.). For all statistical tests, a *p* value of <0.05 was considered as significant.

## 3. Results

### 3.1. Infection and Morbidity

Eight calves were treated throughout the duration of the experiment for haemorrhage or infection. Haemorrhage was identified as the persistence of bleeding in the subsequent days after treatment. Haemorrhage was treated with topical anaesthetic Tri-Solfen (Dechra Veterinary Products (Australia) Pty. Ltd., Somersby, NSW, Australia), which contains a haemostatic agent. Infection was characterised by foul smell and putative exudate. Calves treated for infection were treated with a topical oxytetracycline antibiotic aerosol (Alamycin Aerosol, Norbrook Laboratories, Tullamarine, VIC, Australia). Calves treated for infection on day 13 were additionally administered an intramuscular dose of oxytetracycline (Alamycin Aerosol, Norbrook Laboratories, Tullamarine, VIC, Australia).

One calf (M group) was observed to be slightly bleeding on day 1 and was subsequently treated. This same calf was also treated on day 6 for infection. Two calves from the NC group were treated for infection: one on day 3, 6, and 9 and one on day 9. Five calves from the ML group were treated for infection, three on day 6 and two on day 13.

### 3.2. Blood Analysis

#### 3.2.1. Plasma Meloxicam Concentration

There was a significant interaction between treatment and day for PMC (*p* ≤ 0.001; Figure 1; Table 5). Plasma meloxicam concentration was significantly greater in ML-treated calves than M-treated calves at all timepoints except day 1. PMC was not detectable in M calves on day 0, 9, or 13.

Variation in plasma meloxicam concentration was analysed between individuals within the M and ML treatment groups, finding significant differences between individuals in the ML group only (*p* < 0.001; Table 6; Figure 2).

There was no significant effect of weight on PMC (*p* = 0.845).

#### 3.2.2. Medicated Lick Block Consumption

A total of 15.5 kg of the medicated molasses lick blocks was consumed over the 8 days that the ML group had ad libitum access. An estimate of average meloxicam consumption for each calf is 6.34 mg/kg over the duration of the 8 days, equating to 0.9 mg/kg/day. This is only an estimate, as individual consumption was not measured.

#### 3.2.3. Serum Haptoglobin Concentration

Serum haptoglobin concentration was not normally distributed, and due to the nature of the data, they were unable to be transformed. There was a significant interaction between treatment and day for serum haptoglobin concentration (*p* = 0.041; Figure 2). Serum haptoglobin was significantly lower in PC calves on day 1 and 3. There was no significant difference between M and ML calves at any time point. There was no significant difference between treatment groups by day 13.

### 3.3. Wound Morphology

#### 3.3.1. Wound Healing

Wound scores were first compared between the two different methods used, finding that calves castrated by the initial method of removing the distal portion of the scrotum were significantly more likely to have higher wound scores compared to calves castrated by the latter method of two lateral incisions on either side of the scrotum (*p* = 0.012). The calves castrated with the initial method were subsequently removed from the data for all wound measurements including wound healing, scrotal diameter, and scrotal temperature.

Wound scores between observers were then analysed to determine any variation, finding a significant difference (*p* < 0.0001). Observer III was 4.41 and 2.89 times more likely to assign a greater wound score than observer I and observer II, respectively. Observer III’s scores were subsequently removed from the analysis. There was no significant difference between observer I and II (*p* = 0.1081). The wound healing scores recorded over the experiment were analysed between each group of castrated calves (Figure 3).

A significant interaction between treatment and day was observed for wound healing scores (*p* < 0.001). Wound healing scores were significantly greater in NC calves compared to ML calves at all timepoints (*p* = 0.0199). There was a statistical trend for M wound scores to differ from NC scores (*p* = 0.0614).

There was no significant effect of PMC on the wound scores of ML calves (*p* = 0.460). However, there was a significant effect of PMC on the wound scores of M calves (*p* = 0.003), with higher meloxicam concentration correlating to higher wound scores (Table 7).

There was a significant effect of SHC on wound score, with higher haptoglobin concentrations linked to higher wound scores (*p* < 0.001; Table 8).

#### 3.3.2. Scrotal Diameter

One datapoint was removed from the scrotal temperature data, as the thermograph was not positioned correctly to capture the entire scrotum. There was a significant effect of day on the change in scrotal diameter between each measurement day and day 1 (*p* = 0.015; Figure 4).

Scrotal diameter measurements taken on day 1 indicated a significant difference between treatments (*p* = 0.013). Scrotal diameter of NC calves was significantly smaller than M calves. However, with the scrotal diameter measurements of the five calves castrated by removing the distal portion of the scrotum removed, there was no significant difference between treatment groups for scrotal diameter on day 1 (*p* = 0.132).

#### 3.3.3. Scrotal Temperature

There was no significant effect of treatment or day on scrotal temperature (*p* = 0.927 and *p* = 0.139, respectively). There was no significant interaction between PMC and scrotal temperature (*p* = 0.0812).

### 3.4. Behaviour

#### 3.4.1. Eating

There was a significant interaction between treatment and time for the frequency of observed eating behaviour (*p* < 0.001; Figure 5; Table 9).

#### 3.4.2. Positional and Locomotive Behaviours

There was a significant effect of treatment on the frequency of observed lying behaviour (*p* < 0.001) and locomotive behaviour (*p* = 0.032; Table 10). Lying behaviour was observed most in PC calves and least in ML calves, while locomotive behaviours were observed most in ML calves and least in PC calves.

## 4. Discussion

This study was the first to evaluate the efficacy of the self-administration of meloxicam via medicated lick blocks when provided ad libitum to surgically castrated calves. Physiological and behavioural responses to surgical castration were evaluated to determine the efficacy of meloxicam delivered via a medicated lick block compared to conventional subcutaneous injection. Presumed therapeutic concentrations of meloxicam were established and maintained for ten days, with the medicated lick block removed after seven days of ad libitum provision. Wound healing scores were lower (more healed) in meloxicam-treated calves compared to negative control calves. Eating behaviour was increased in ML calves immediately after castration compared to all other treatment groups and was reduced after removal of the medicated lick block. Locomotion was greater in ML calves compared to all other treatment groups. Behavioural and physiological data indicate that the administration of meloxicam via medicated lick blocks is effective for providing pre-emptive and long-term analgesia in surgically castrated calves.

The successful administration of meloxicam via a medicated lick block enabled therapeutic concentrations of the drug to be established pre-emptively and sustained long-term, without human interference. The therapeutic concentration of meloxicam for cattle is estimated from the half-maximal effective concentration (EC_50_) of meloxicam in equines and canines: 0.73 µg/mL and 0.36 µg/mL, respectively [21,36,37,38]. The predicted mean plasma meloxicam concentration (PMC) for ML calves at the time of castration, after 19 h of ad libitum access, was 0.64 µg/mL. A study comparing the pharmacokinetics of subcutaneous and oral meloxicam found that the time to establish maximum drug plasma concentration (T_max_) was 3.7 and 24 h, respectively [15]. These results indicate that although presumed therapeutic concentration was established by the time of castration, providing the medicated lick blocks for 24 h pre-operative may have increased PMC and therefore the therapeutic effect [22]. However, the average daily PMCs in the ML group (3.16 µg/mL) were similar to, and generally higher than, concentrations reported in previous studies that have investigated the pharmacokinetics and efficacy of oral meloxicam in cattle [39,40]. In such studies, oral meloxicam was administered once to cattle at a dose rate of 0.5 mg/kg body weight [40] or 1 mg/kg body weight [39], and was shown to be effective at relieving pain caused by experimentally induced lameness [40] and cautery dehorning [39], respectively.

The route of administration impacts the rate of absorption of meloxicam, and therefore the onset and duration of efficacy. Subcutaneous meloxicam delivered at a rate of 0.5 mg/kg typically establishes therapeutic concentration in significantly less time but maintains therapeutic concentrations for a significantly shorter time compared to oral meloxicam administered at a dose of 1.0 mg/kg [15,21]. The average PMC of M calves one day post-administration was 1.33 µg/mL, which was not significantly different to the average PMC of ML calves on that day. This suggests that the concentration of meloxicam in the lick blocks relative to lick block consumption is approximate to the dose for oral administration. The PMC on the subsequent days for M calves was significantly reduced each day, being almost undetectable by day 6. Subcutaneous meloxicam has been shown to reduce behavioural and physiological indicators of pain associated with surgical castration in calves for up to 72 h [6]. The predicted average PMC of M calves on day 3 was (0.28 ± 1.32 µg/mL), slightly lower than the presumed therapeutic plasma concentration previously discussed [21,36,37,38]. The average PMC of ML calves peaked on day 2 (5.59 ± 1.32 µg/mL) and remained elevated with a maximum individual concentration of 17.41 µg/mL detected on day 6. The average PMC of ML calves on day 9 of the study, three days following removal of the medicated lick block on day 6 was 0.87 µg/mL, still within presumed therapeutic range [36,37,38]. The elevated concentration and maintenance of therapeutic concentration after lick block removal can be attributed to the long elimination half-life and slower absorption of oral meloxicam [20,21,23].

The recommended dose of oral meloxicam in cattle is 1.0 mg/kg (Solvet, Alberta Veterinary Laboratories, Calgary, AB, Canada), compared to 0.5 mg/kg OTM and SC meloxicam (Boehringer Ingelheim Vetmedica, Rohrdorf, Germany). Over the seven days of ad libitum access to the medicated lick block, each calf consumed an average estimate of 0.9 mg/kg/day, based on the overall consumption of the block and group liveweight. However, individual PMC analysis highlighted significant differences in meloxicam intake between calves within the ML group, with concentrations ranging from 0.59 to 17.41 µg/mL on the same day (day 6). The palatability of molasses lick blocks medicated with meloxicam had been previously analysed by comparing the consumption of medicated and non-medicated lick blocks in a group of Holstein heifer calves, finding that the consumption of both blocks was similar [41]. Despite this, it is possible that some calves were aversive to the lick blocks where others were drawn to them, be it due to taste or seeking nutritional supplementation [42]. It was expected if any aggression or bullying would occur that the larger calves would be dominant over the smaller calves; however, there was no correlation between weight and PMC in ML calves. Ultimately, individual variation in PMC in ML calves was likely just a product of varied intake. In a commercial setting, individual lick block intake would seldom be monitored or controlled. This methodology demonstrates the most practical on-farm application, with the lowest PMC values on day 2 to 6 showing established estimated effective concentration. However, the high maximum values have potential implications for meloxicam toxicity.

There was a significant difference in haptoglobin concentration between sham castrated calves and castrated calves on days 1 and 3, where haptoglobin levels in PC calves were elevated, only returning to near-undetectable levels by day 9. These results indicate that serum haptoglobin concentration (SHC) may be influenced by stress alone. Similar outcomes have been seen in other studies where non-castrated and non-dehorned control calves’ haptoglobin levels did not differ to their treated counterparts on the day of castration/dehorning [15,39,43]. Additionally, the stress involved in transportation, isolation, and environmental changes [44], weaning [45] in mature cattle, and different transportation conditions for calves have been shown to increase haptoglobin concentrations [46]. Haptoglobin is an acute-phase glycoprotein that is released into the bloodstream in response to tissue injury such as trauma or infection, having an important interaction with haemoglobin [47,48]. Haptoglobin concentration typically increases during instances of acute inflammation, potentially providing information about the severity of tissue damage and the effect of anti-inflammatory measures [47,49,50]. Although being near-undetectable in the healthy animal, haptoglobin is a major acute phase protein in cattle, possibly increasing by a factor of 100 in instances of acute inflammation, making it an effective indicator of inflammation [45,51].

There was no significant difference in SHC between castrated calves except for day 6, where NC calves had significantly higher haptoglobin levels than M calves, while there was no difference between ML calves and other castrated calves at this time. The anti-inflammatory effect of meloxicam may reduce the production of acute phase proteins such as haptoglobin by inhibiting the biosynthesis of prostaglandin, which has a major role in the initiation of inflammation [52,53]. However, similar to the results seen in the present study, numerous studies have not seen an effect of meloxicam on haptoglobin concentration in various husbandry procedures and stressful events in cattle [15,39,43,54,55,56,57,58] or sheep [59]. It is possible that the haptoglobin concentrations may have been affected by instances of infection. A total of eight calves were treated for infection: one M calf, two NC calves, and five ML calves. Haptoglobin concentration appears to plateau in the ML group from day 6 to 13, corresponding to the days when these calves were treated, potentially indicating the high infection rate. Haptoglobin has been established as a valuable biomarker for various bacterial infections in cattle [60,61,62,63,64,65]. However, without a more comprehensive examination of infection, a definitive connection cannot be made.

There was a significant interaction between treatment and day for wound healing, with NC calves having significantly higher (less healed) wound scores compared to calves receiving meloxicam lick blocks at all timepoints. There was a statistical trend for NC calves to have significantly higher wound scores than subcutaneous-meloxicam-treated calves. These results suggest that wound healing was improved with the administration of meloxicam, which has not previously been demonstrated in the literature. No effect of wound healing has been observed in other studies evaluating the effect of meloxicam on wound healing for surgical castration [31,66], band and surgical castration [28], surgical castration and hot-iron branding [58], hot-iron branding [67] or cautery disbudding [68] in cattle, mulesing in lambs [59], or orthopaedic surgery in horses [69]. A negative effect of meloxicam on healing has been observed in disbudded calves administered a secondary dose of meloxicam 3 days post-operation and initial meloxicam administration [70]. Meloxicam is an anti-inflammatory drug, and inflammation is one of the four non-exclusive stages of healing: haemostasis, inflammation, proliferation, and remodelling [71,72,73]. A review into the effect of meloxicam on wound healing in Murphy Roths large mice, a strain of laboratory mouse with superior healing capabilities, found that meloxicam inhibited wound healing in ear hole closure models [74]. In this study, wound healing was determined purely from visual observation, scored on the closure of the incision, visibility or protrusion of flesh, degree of wound dermatitis, and the presence of exudate, not factoring in any histopathological aspects [31,75]. It is likely that the wounds of meloxicam-treated calves visually appeared more healed due to some reduced inflammation but were delayed in healing on a cellular level.

In all treatment groups, wound healing scores were significantly higher on days 1 and 2 following castration compared to days 3, 6, 9, and 13. Additionally, there was a significant effect PMC on wound scores in M calves. The relationship between PMC and wound healing is more likely a function of time rather than an effect of meloxicam. This is because the highest wound score corresponds to the highest meloxicam concentration, which was on day 1, and reduced with time, corresponding to lower wound scores. To further explore this interaction, a larger sample size being examined for a longer time may be necessary. There was also a significant effect of SHC on wound score. Haptoglobin plays an important role in inflammation, which is critical to wound healing, and is therefore commonly used as a measure of inflammation [47,49,50]. Similar to the PMC data, the positive correlation observed between haptoglobin and wound scores is likely a product of time as there was no treatment effect.

ML calves were observed eating more frequently than all other treatment groups immediately post-castration. Approximately 3 h later, in the afternoon following castration, M calves displayed significantly more eating behaviour than all other treatment groups. The increase in eating for M calves corresponds with the T_max_ of SC meloxicam, which is approximately 3.7 h, highlighting the delayed onset of therapeutic effect of peri-operative administration [15]. Cattle have been observed to spend up to 13 h a day eating, with the rest of the day mostly comprising ruminating and resting [76]. Therefore, any marked reduction in eating behaviour can be considered as abnormal [77]. These results contrast results of a similar study comparing timing of administration of meloxicam at 6 or 3 h pre-castration and immediately prior to castration, which found no treatment effect for multiple eating parameters [54]. Potentially confounding these results is the significantly lower instance of eating in PC calves at both timepoints. The PC group was processed after the NC group, prior to the M and ML group, administered a standing sedation only at this time. The calves were held in the yards off feed for approximately 3.5 h during castration on day 0, so it is very curious that eating behaviour was so limited in PC calves at this time. These results contrast with similar studies comparing the effect of castration on behaviour, finding that eating behaviours occurred significantly more in non-castrated controls compared to castrated calves [78] and concurrently castrated and dehorned calves [79].

There was no discernible pattern in eating behaviour over the duration of the experiment. Eating behaviour declined significantly in ML calves from day 6 to day 9 and 13, corresponding to the removal of the medicated lick blocks on day 6. However, the individual variation in PMC between ML calves complicates connecting these two factors. Eating behaviour from day 6 to day 13 was reduced in all treatment groups, indicating a potential environmental influence. Each treatment group was separately held in the same 0.2 ha yard for the duration of the trial, with ad libitum access to hay, molasses lick blocks, and water. It is likely that over 14 consecutive days in the same area, the pasture quality had diminished, leading to reduced eating behaviour. Additionally, feeding behaviour in cattle in feedlot and pasture conditions tends to peak around dawn and dusk [76,80]. Considering this, the timing of behavioural observations during this experiment may have merely missed peak grazing activity.

A significant effect of treatment was observed for lying and locomotion. Positional and locomotive behaviours can inform observers about comfort or pain experienced by cattle [29,81]. Lying was seen significantly more in PC calves compared to all other treatment groups. A positive effect of meloxicam on lying behaviour has been seen in numerous studies looking at cautery disbudding [82,83], amputation dehorning [36], concurrent surgical castration and amputation dehorning when combined with topical anaesthetic [79], and caesarean section [84]. However, several studies evaluating the effect of surgical castration have found no effect of NSAIDs flunixin meglumine [34,75] or meloxicam [54,85] on lying behaviour after surgical castration. Concurrently, as ML calves were observed lying significantly less, they performed significantly more locomotive behaviours compared to all other treatment groups. Several studies evaluating the effect of surgical castration on the activity levels of calves have found that activity is significantly reduced following the procedure compared to pre-surgery activity [86] or compared to non-castrated counterparts [78]. There are conflicting reports in the literature about the effect of meloxicam on locomotion, with several studies finding no effect on locomotion [15,27,85]. However, some studies have found that locomotion is increased in calves administered meloxicam for surgical castration compared to calves administered placebo [6] and a combination of meloxicam and topical anaesthetic for concurrent surgical castration and amputation dehorning [79] compared to non-castrated control calves. Locomotion is commonly measured using accelerometry rather than visual observation, as it allows for the continuous remote monitoring of both dynamic and static movement [86]. In the present study and the study by Van der Saag et al [79], locomotion was measured only by visual observation. Similar to the eating behaviour data, observations were only made at specific timepoints, rather than continuously, limiting the interpretation of results. Ultimately, the lack of clear patterns in behavioural responses observed in the present study highlight the complexities of behavioural measurement of pain and interpretation of results in animals. Multiple studies have observed variation in behaviour between individuals in response to painful procedures [79,87].

No significant treatment effect for scrotal swelling between castrated calves were observed. These findings are consistent with some other studies examining scrotal swelling of surgically castrated calves administered meloxicam [31,58] or flunixin [75]. Oedema is the accumulation of extracellular fluid caused by increased vascular permeability or inadequate lymphatic drainage and generally subsides after approximately 14 days [88,89,90]. Scrotal oedema can be measured as an indication of inflammation and healing in surgically castrated calves by measuring scrotal diameter [6,31]. As there was a significant difference between treatments for weight, it was posited that this effect may extend to scrotal diameter due to differences in development. To eliminate this potential effect, the difference between day 1 scrotal measurements and those of each subsequent day was compared. The administration of meloxicam via oral suspension has been shown to reduce scrotal swelling in surgically castrated calves over the first 3 days post-operation [6]. In the studies that saw no effect of NSAID administration on scrotal swelling, measurements were taken while calves were in lateral recumbency, whereas in the present study and Olson et al.’s study, scrotal swelling was measured while calves were standing. These key differences in methodology may have led to the concentration of oedema in the distal portion of the scrotum of standing calves due to gravity [31]. Despite this, no treatment effect was seen in the present study, which is likely a result of the small sample size with the additional removal of data from five NC calves. These results also indicate that scrotal diameter may not be a very direct measure of inflammation.

There was no effect of treatment or day on scrotal temperature in this study. These results contrast to other studies evaluating the effect of meloxicam on surgical castration, finding that scrotal temperature was significantly reduced over the first 2 days post-castration [31] and over days 1–7 post-castration [85]. The sample sizes and data collection timepoints were very similar between these previous studies and the present study. Increased wound temperature can indicate the presence of inflammation and can be measured non-invasively through the use of infrared thermography [91,92,93]. The increase in wound temperature during acute inflammation can be attributed to the rapid influx of leukocytes and proliferation of macrophages [52]. A study evaluating environmental influences on scrotal temperature in bulls found that scrotal temperature was significantly increased by feeding for up to 3 h, lying (for a minimum of an hour) during high and low temperatures (25 °C and 5 °C), and ambient temperature (at the distal portion of the scrotum), and was significantly reduced by moisture on the scrotum [94]. In the present study, the entire scrotum was captured in the infrared image, with the software program identifying the maximum temperature throughout the entire image, which was used for data analysis. These outcomes suggest that scrotal temperature in the present studies may have been influenced by environmental factors, possibly accounting for the differences between studies of similar experimental design. Similar to scrotal diameter, scrotal temperature may not be a great direct measure of inflammation.

A major limitation of this study was the removal of wound scores and scrotal diameter and temperature data from the five NC calves that were castrated using a different method. Significant differences were identified in wound scores between the two different castration methods within the NC group. Subsequently, the data from these calves were also removed from other wound-based measurements to ensure reliability. This diminished the statistical power of the experiment. The healing of surgical castration wounds has been shown to take approximately 4 to 9 weeks [75]. Considering this, a longer observational period may be required to observe the complete healing process to fully understand the effect of continued administration of meloxicam on wound inflammation and healing. The random allocation of calves to treatment groups led to unequal variance in weight between treatment groups. In future, calves should be blocked by weight and/or scrotal circumference. Additionally, calves should be treated randomly on day 0 to alleviate behavioural differences in the immediate post-sedation period. Individual lick intake in the ML group was not measured, making it impossible to truly understand the variation in PMC. Future research in this area should consider this to determine the factors that influence intake. The positive control group were steers that had been castrated within a week after birth, therefore not representing a true sham castration group. However, in the context of this study, the inclusion of steers did not limit the statistical power of any wound-related measures, as comparison to intact bulls would not have been relevant. A baseline scrotal temperature may have been advantageous to compare increases in temperature due to inflammation and may have identified any natural individual variation in scrotal temperature.

Toxicity to continuously administered meloxicam has been reported in numerous species including sheep, dogs, rats, and horses [95,96,97,98], but has not been reported for cattle. Meloxicam is often the preferred NSAID for animal treatment due to the long duration of efficacy and COX-2 specificity for the inhibition of prostaglandin [99,100]. COX-2 is preferred over COX-1, as it is mainly expressed at sites of inflammation; however, COX-2 also plays an important role in production of prostaglandin for homeostasis of the central nervous system, renal system, gastrointestinal system, bones, female reproductive tract, and cardiovascular system [96,101,102,103]. Meloxicam toxicity often manifests as gastrointestinal ulceration or perforation and impaired renal function in monogastric species [95,96,97,99]. Experimentally induced meloxicam toxicity in rats at a dose of 2.3 mg/kg/day for 28 days led to renal, hepatic, myocardial, and neuronal necrosis [97]. In horses, a dose of 0.6 mg/kg PO meloxicam every 24 h for 6 weeks was tolerated well with no adverse effects, whereas horses that were administered 3 and 5 times this dose experienced gastrointestinal damage, renal damage, and bone marrow abnormalities [98]. The administration of a single dose of oral meloxicam at a rate 30 times the recommended rate in dairy cattle was evaluated, finding no adverse effects or histological abnormalities of the kidneys, liver, or gastrointestinal tract 10 days post-treatment [104]. Although, the administration of NSAID ibuprofen to 5–6-week-old calves at a rate of 30 mg/kg fed orally via milk replacer three times daily (3 times the recommended dose) for 10 days resulted in the development of abomasal ulcers in five out of eight calves [105,106]. Similarly, the administration of meloxicam at a rate of 30 mg/kg for three consecutive days to lactating sows saw the development of subacute gastric ulcers in 2/5 sows and in 10/11 piglets suckling off of these two sows [107]. In addition to toxicity, the withholding period of continuously administered meloxicam needs to be determined. These findings indicate that research into the toxicity effects of continuous meloxicam administration in cattle is critical for future research in this field.

## 5. Conclusions

The self-administration of meloxicam by calves provided with ad libitum access to medicated molasses lick blocks successfully enabled pre-emptive and continuous administration of meloxicam. While there was great variation in plasma meloxicam concentrations within the medicated lick block group, several calves were able to establish and maintain presumed therapeutic concentrations of the drug. The administration of meloxicam, both subcutaneously and via medicated lick blocks, appeared to improve wound healing and the instances of abnormal behaviour in the weeks following surgical castration. One major limitation of this study is the lack of measuring individual lick intake in the medicated lick block group, inhibiting the determination of the cause of individual variation in plasma meloxicam concentration. Another major limitation is the change in castration method, which occurred halfway through castrating the negative control group. This was carried out in the best interest of animal welfare; however, it limited the power of statistical analysis of wound parameters. Additionally, the unclear patterns in behavioural data highlight the complexities of measuring pain in animals Ultimately, future projects in this space need to consider these limitations and additionally evaluate the potential toxicity effects from continued administration of meloxicam. Although this study highlights the complexities of assessing pain in cattle, there were some indications for improved welfare of surgically castrated calves following self-administration of meloxicam through a medicated molasses lick block.

## Figures and Tables

**Figure 1 animals-15-00442-f001:**
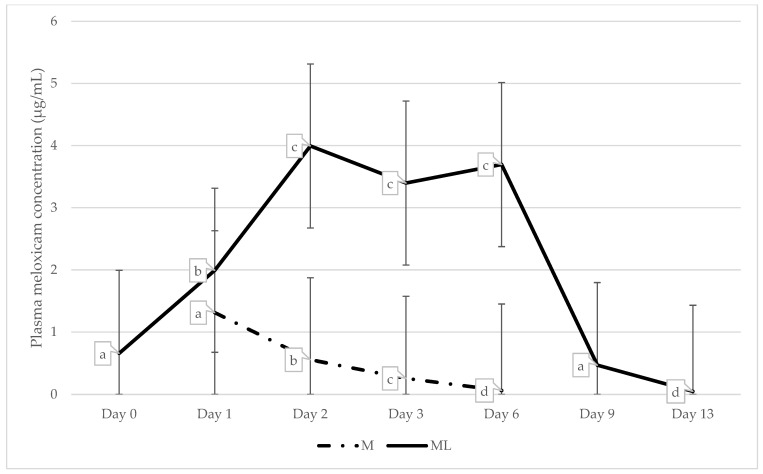
Predicted means and standard error of the means (s.e.m.) for plasma meloxicam concentration (µg/mL) of calves in treatment groups: M = surgically castrated with subcutaneous meloxicam; ML = surgically castrated with access to meloxicam-medicated lick blocks. Values with different superscripts differ significantly (*p* ≤ 0.001). Different values ^a,b,c,d^ represent significant differences across days within treatment groups.

**Figure 2 animals-15-00442-f002:**
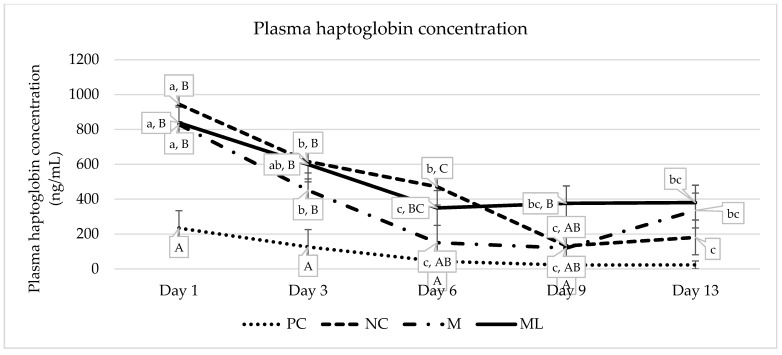
Serum haptoglobin concentrations (ng/mL) and standard error of the means between all treatment groups from day 1 to day 13. Values with different superscripts differ significantly (*p* ≤ 0.041). Lowercase values ^a,b,c^ represent significance within treatments over the timepoints, and uppercase values ^A,B,C^ represent significant difference between treatments groups within a timepoint. PC = non-castrated positive control; NC = surgically castrated; M = surgically castrated with subcutaneous meloxicam; ML = surgically castrated with access to meloxicam-medicated lick blocks.

**Figure 3 animals-15-00442-f003:**
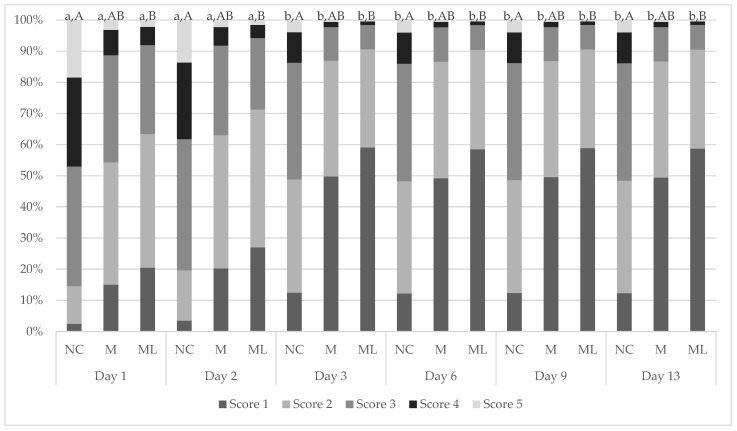
Predicted proportions of wound healing scores between each treatment group on each day of measurement. Different values ^a,b^ indicate significant differences within a treatment between days (*p* < 0.05). Different values ^A,B^ indicate significant differences between treatments within a day. NC = surgically castrated; M = surgically castrated with subcutaneous meloxicam; ML = surgically castrated with access to meloxicam-medicated lick blocks.

**Figure 4 animals-15-00442-f004:**
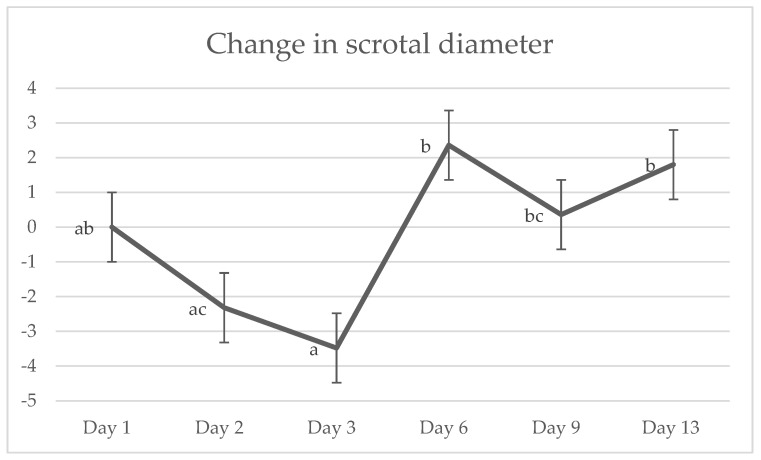
The predicted means for change in scrotal diameter between each day compared to day 1. Data points with different values ^a,b,c^ differ significantly (*p* = 0.015). Data points with no value were not significant to any other datapoint.

**Figure 5 animals-15-00442-f005:**
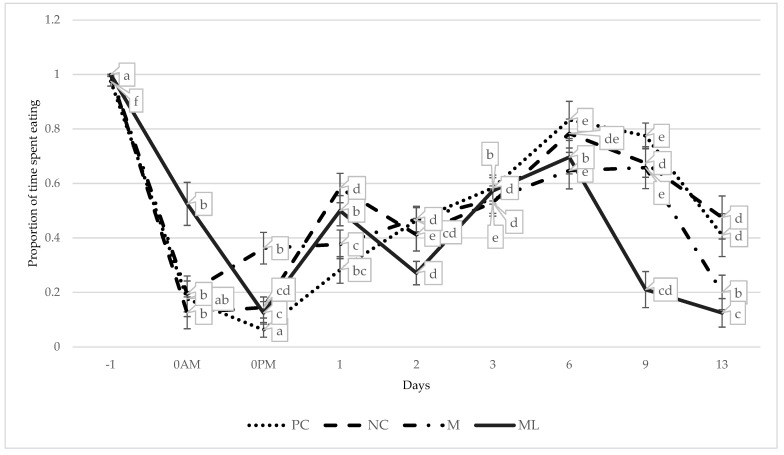
The predicted proportions of time spent eating throughout the experiment for each treatment group. Datapoints with different values ^a,b,c,d,e,f^ indicate values differing significantly across time within a treatment group (*p* < 0.001). PC = non-castrated positive control; NC = surgically castrated; M = surgically castrated with subcutaneous meloxicam; ML = surgically castrated with access to meloxicam-medicated lick blocks.

**Table 1 animals-15-00442-t001:** Sedation and castration times per treatment group on day 0.

Treatment Group	Sedation Time	Castration Time
PC	09:20–09:37	-
NC	08:30–09:03	08:35–09:29
M	09:44–09:54	09:52–10:32
ML	10:23–10:37	10:42–11:34

**Table 2 animals-15-00442-t002:** Lick block consumption by ML calves via weight analysis.

Day	Lick Block Weight (kg)	Change in Weight (kg)	Meloxicam Consumption (mg)
−1	39.85	0	0
0	38.5	1.35	1350
1	37.4	1.1	1100
2	35.25	2.15	2150
3	33.15	2.1	2100
6	24.35	8.8	8800
Total		15.5	15,500

**Table 3 animals-15-00442-t003:** Wound morphology scoring scale.

Score	Example	Description
**1**	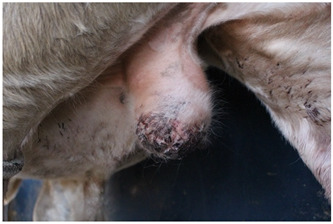	Complete closure of the incision Mild wound dermatitis No exudate No exposed tissue
**2**	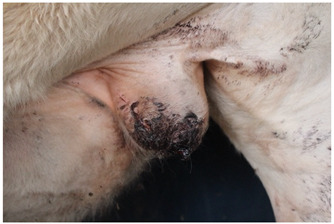	Incomplete incision closure Mild wound dermatitis No exudate Very small amount of exposed tissue
**3**	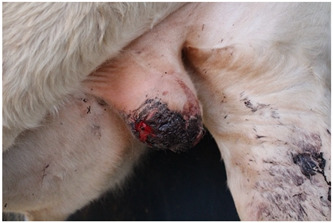	Incomplete incision closure Mild to moderate wound dermatitis Minimal exudate, visible tissue is dry Tissue is visible, does not protrude wound
**4**	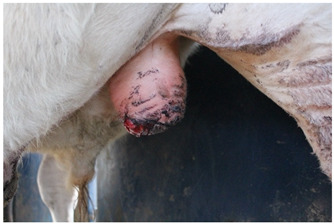	Incomplete closure of the incision Mild to moderate wound dermatitis Minimal exudate, visible flesh is moist Tissue is visible, flesh protrudes wound
**5**	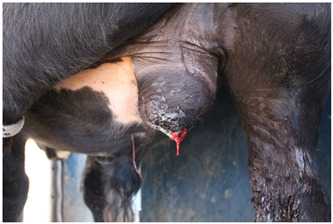	Incomplete closure of the incision Moderate wound dermatitis Exudate from wound, may still be bleeding or clotted Tissue may be visible and protrude flesh.

**Table 4 animals-15-00442-t004:** Ethogram of behaviour. The ethogram was adapted from validated ethograms in the literature [13,29,31,32,33,34,35].

Behaviour	Description
Position/Locomotion	
Normal standing	Calf stands passively or actively with muscles relaxed, head may be held in relaxed frame, ears may be held loose
Abnormal standing	Calf stands stationary, possibly hunched back, legs may be trembling or stretched out, body may be trembling, may be shifting weight between hind legs (easing quarters), head may be held low
Normal lying	Calf may lie ventrally or laterally, legs may be folded underneath or partially, head may be up or folded towards body
Abnormal lying	Calf lies laterally with hind legs entirely or partially extended, lateral recumbency, forelimbs may be extended, head may be flat on the ground
Normal moving	Calf moves in forward locomotion, takes even steps, weight bearing on all 4 limbs, even cadence, relaxed muscles, hindfoot replaces forefoot
Abnormal moving	Calf may walk with a hunched back, trembling legs, small, short steps, tiptoeing, walking on knees, falling over, stiff gait, spreading of hind legs or dragging feet, hindfoot may not replace forefoot
Activity	
Tail flicking	Calf flicks its tail in one motion or one continuous motion
Ear flicking	Calf flicks one or both ears
Kicking or stamping	Calf may kick at belly and/or/groin, may kick backwards, stomping fore or hind leg, definitive lifting and replacing of the foot not associated with walking
Head turning	Calf turns its head to side of body, may place head under leg towards wound
Vocalisation	Calf vocalises
Grooming	Calf licks self
Maintenance	
Grazing	Calf grazes, ingests feed—pasture, hay or lick block
Drinking	Calf ingests water
Defecation/urination	Calf defecates or urinates
Ruminating	Calf chews cud, salivates

**Table 5 animals-15-00442-t005:** The variation in PMC (µg/mL) between M and ML calves described by the means, standard deviations, minimum and maximum values per day.

	M				ML			
	Mean	SD	Min	Max	Mean	SD	Min	Max
Day 0	−				1.06	1.13	0.24	3.47
Day 1	1.33	0.26	1.03	1.90	3.64	3.35	0.11	9.81
Day 2	0.58	0.18	0.42	1.02	5.59	4.04	0.82	12.06
Day 3	0.28	0.13	0.10	0.58	4.70	3.38	0.52	10.66
Day 6	0.08 *	0.04 *	0.05 *	0.14 *	6.47	6.18	0.59	17.41
Day 9	−				0.87 ***	0.74 ***	0.09 ***	2.34 ***
Day 13	−				0.13 **	0.08 **	0.05 **	0.25 **
Average	0.57	0.15	0.27	0.61	3.21	2.70	0.35	8.00

− = meloxicam concentration was either not detectable or was below LOQ < 0.0048 µg/mL. * *n* = 5 (5 excluded as LOQ < 0.048 µ/mL); ** *n* = 8 (2 excluded as LOQ < 0.0048 µg/mL); *** *n* = 9 (1 excluded as LOQ < 0.0048 µg/mL).

**Table 6 animals-15-00442-t006:** Predicted mean plasma meloxicam concentrations (µg/mL) ± standard error of the mean (s.e.m) of individual calves in the medicated lick block (ML) group. Values with different superscripts differ significantly (*p* ≤ 0.001).

Calf	PMC (µg/mL) ± s.e.m
31	1.49 ^a^ ± 1.96
32	0.20 ^b^ ± 2.06
33	3.30 ^cd^ ± 1.96
34	3.61 ^c^ ± 1.96
35	2.04 ^ae^ ± 1.96
36	1.02 ^ab^ ± 1.98
37	2.46 ^de^ ± 1.98
38	0.49 ^b^ ± 1.98
39	0.30 ^b^ ± 1.98
40	0.64 ^ab^ ± 1.98

**Table 7 animals-15-00442-t007:** The predicted mean plasma meloxicam concentration (µg/mL) ± standard error of the mean (s.e.m) of subcutaneous meloxicam (M)-treated calves for each wound score.

Wound Score	PMC ± s.e.m.
**1**	0.17 ^a^ ± 1.23
**2**	0.21 ^ab^ ± 1.23
**3**	0.48 ^c^ ± 1.29
**4**	0.69 ^bc^ ± 1.71
**5**	1.16 ^c^ ± 1.80

^a,b,c^ indicate significant differences between wound scores.

**Table 8 animals-15-00442-t008:** Predicted means of serum haptoglobin concentration (SHC; ng/mL) ± standard error of the mean (s.e.m) of all castrated calves for each wound score.

Wound Score	SHC ± s.e.m.
**1**	290 ^a^ ± 47.3
**2**	423.1 ^b^ ± 53.2
**3**	602 ^c^ ± 59.4
**4**	613.3 ^cd^ ± 84.1
**5**	816.7 ^d^ ± 89.8

^a,b,c,d^ indicate significant differences between wound scores.

**Table 9 animals-15-00442-t009:** The frequency of eating within an approximately 2 h period was used to predict the proportion of time spent eating for each treatment group by day. ^A,B^ indicate differences between treatments within a timepoint (*p* < 0.001). PC = non-castrated positive control; NC = surgically castrated; M = surgically castrated with subcutaneous meloxicam; ML = surgically castrated with access to meloxicam-medicated lick blocks.

DAY	PC	NC	M	ML
−1	0.98 ± 0.02	1.00 ± 0.00	1.00 ± 0.00	1.00 ± 0.00
0AM	0.18 ^A^ ± 0.07	0.12 ^A^ ± 0.06	0.19 ^A^ ± 0.07	0.53 ^B^ ± 0.08
0PM	0.06 ^A^ ± 0.03	0.14 ^A^ ± 0.04	0.36 ^B^ ± 0.06	0.13 ^A^ ± 0.04
1	0.28 ^A^ ± 0.05	0.58 ^B^ ± 0.05	0.38 ^AB^ ± 0.05	0.5 ^B^ ± 0.06
2	0.46 ^A^ ± 0.05	0.41 ^AB^ ± 0.06	0.47 ^A^ ± 0.05	0.27 ^B^ ± 0.04
3	0.58 ± 0.05	0.53 ± 0.05	0.54 ± 0.05	0.57 ± 0.05
6	0.83 ± 0.07	0.78 ± 0.05	0.65 ± 0.07	0.70 ± 0.06
9	0.78 ^A^ ± 0.05	0.68 ^A^ ± 0.05	0.66 ^A^ ± 0.08	0.21 ^B^ ± 0.07
13	0.41 ^A^ ± 0.08	0.48 ^A^ ± 0.08	0.2 ^B^ ± 0.06	0.13 ^B^ ± 0.05

**Table 10 animals-15-00442-t010:** Predicted proportions ± standard error of the means of time spent lying or locomoting between treatment groups. Treatment groups: PC = non-castrated positive control; NC = surgically castrated; M = surgically castrated with subcutaneous meloxicam injection at time of castration; and ML = surgically castrated with access to meloxicam-medicated lick blocks. Values with different superscripts ^a,b,c^ differ significantly (*p* < 0.05).

	Lying *p* = <0.001	Locomotion *p* = 0.032
PC	0.24 ^a^ ± 0.02	0.08 ^a^ ± 0.01
NC	0.18 ^b^ ± 0.01	0.09 ^a^ ± 0.01
M	0.19 ^b^ ± 0.02	0.10 ^a^ ± 0.01
ML	0.11 ^c^ ± 0.01	0.13 ^b^ ± 0.01

## Data Availability

Data are available on request from the authors.

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
