# Peer review of "Self-Administration of Meloxicam via Medicated Molasses Lick Blocks May Improve Welfare of Castrated Calves"

_animals, 2025, doi:10.3390/ani15030442_

Round 1

Reviewer 1 Report

Comments and Suggestions for Authors

The authors have assessed the effects of meloxicam administered through the conventional SC route and through an innovative self-medication via molasses lick block on post-castration parameters in 4- to 6-month-old calves. They have demonstrated that the later means of administering meloxicam yielded plasma drug concentrations putative of inducing clinical effects in calves, which could be of major clinical relevance. However, the paper itself needs to be improved and I have provided various comments below.

Although the experimental design seems to be appropriate, it could have been improved. For instance, the need of including a surgical castration (NC) group is questionable. This is considering the vast information on behavioral and wound changes associated with castration in calves. Including such a group seems more of a potential welfare concern than a valid refinement of the study design. Having a true sham castrated group, a positive control group (SC meloxicam) and a test group (meloxicam in molasses lick blocks) would have sufficed. Furthermore, data from only 5 animals in the NC group were analyzed due to differences in castration techniques and, as discussed (e.g., line 493), this group was likely underpowered to provide any meaningful comparisons. The authors should consider whether to remove this group completely from the analysis. Also, the use of steers as a “no castration” group (line 110) seems inappropriate. Using bull calves in the PC group, and thus having a true sham castration group, would have been a much better option. This, at least, should have been discussed as a limitation of the study.

As recognized by the authors, pain assessment in animals is challenging. Proxy measures such as animal production outcomes (e.g., liveweight) are not specifically indicative of pain. However, evaluating changes of species- and pain-specific behaviors using validated instruments is one of the most robust ways to assess pain in animals. Validated cattle pain scales specific for castration are available and the authors are aware of them (reference 81; see also Animals 13: 364, 2023; and Sci Rep 14: 15257, 2024). Constructing and using a bespoke ethogram without validating it prior to performing the study is a major drawback. What was the reasoning of using this ethogram over a validated species- and pain-specific scale? This, at least, should be discussed.

In a similar way, the authors are aware of the large number of animals and long time needed to detect differences in liveweight (lines 507-509). Despite that, this parameter was measured without even blocking for this parameter wen assigning animals to the different groups. Unsurprisingly, no significant differences were found between treatment groups of 5-10 animals during the 2-week duration study. Discussing liveweight (lines 488-515) seems meaningless and deters from what I believe it should be the focus of the paper (i.e., delivery of meloxicam in the molasses lick blocks).

Please clarify based on what a sample size of 10 animals per group was selected. Was an a priori power analysis performed? If so, please provide details.

The reasoning for using digital photographs to assess wound healing instead of real-time assessments needs to be clarified. Details about how the photographs were acquired and their quality, and the resolution of the screens used for displaying the photographs for assessment are required. Also, the training/experience of each of the 3 scorers with post-castration assessment in calves needs to be described. Was a formal training session provided, ideally before starting the trial, to the 3 scorers to homogenate assessments? Clarify whether wound scorers were blinded to treatments.

More details about the molasses lick blocks used in the study are also necessary. Please describe the weight of each block and how many blocks were available in each pen. Other than at day 6 for the ML group, were blocks replaced during the study? The way that the meloxicam was incorporated in the molasses lick blocks is also relevant as it is the quality control data for each of the blocks. Was there any testing performed to confirm that for each kg of molasses lick block 1 g of meloxicam was contained and that the NSAID was evenly distributed throughout the lick block? Also, why was the weight of the molasses lick blocks measured only for the ML group but not for the other groups? This even more considering that differences in body weight were assessed.

Details about the performance in your lab for the HPLC assay to determine meloxicam and the immunoperoxidase assay to determine haptoglobin are needed. These should include limits of detection and quantification, and coefficients of variation.

For clarity, please include and match the combined behaviors described in lines 279-287 with those detailed in Table 2. Also, please explain how is it that eating behavior was ‘amalgamated’ with other ‘normal’ behaviors (line 286) and it was then presented separately from any other behaviors (lines 442-452). Also, why data for eating on day -1 (line 232) were not presented in Figure 8 or Table 6? Consider deleting Figure 8; the same data are depicted, in a much tidier way, in Table 6. If the latter suggestion is followed, incorporated lower case superscripts to Table 6.

Please indicate the breed of the animals used in the study as well as the source of the animals. Were the calves breed on the University farm or were bought somewhere and transported to the University farm?

It seems that some parameters were not measured before castration at day -1 (e.g., scrotal diameter, would temperature). When comparing the effects of treatment over time, having baseline measurements is appropriate. This would hopefully provide evidence that all groups are similar to each other before starting the study. Thus, any differences after treatment could be ruled out from groups being different to start with. This at least should be discussed as a limitation of the study.

Consider including a timeline of procedures and measurements to summarize the events the animals underwent during the study. This should include the time that the first and last calf were castrated/‘sham’castrated.

The discussion is too long and should be more focus on the main findings of the study. Some information does not add anything to the discussion of your results (e.g., lines 573-580, 604-607, 791-809) or repeats what it has been stated somewhere else (lines 610-617) and could be deleted. Although some limitations of the study are discussed, other important ones are missing and should be included. For instance, no true sham castration group, the effect of high infection rate (27%; 8 out of 30 calves) post-castration on your results, the age of the calves used and the potential different results in younger animals (J Vet Pharmacol Therap 47 (2): 143-149), the lack of technological aids (sensors) to assess activity, including grazing and ruminating activity, etc.

There is an excessive use of references and not all of them are necessary (e.g., for the same concept two references are provided in line 68 and then two other references in line 794; for discussing haptoglobin results in lines 573-602, 21 new references were cited). Primary sources of information should be cited rather than works citing the primary sources (e.g., in lines 521 and 547, references 18 and 32 used cows and calves, not horses or dogs). The need of citing more than 1 reference for basic concepts (e.g., line 577) should be avoided.

Consider using a reference manager software and make sure that all cited references are formatted according to the journal specifications. For instance, references 4, 6, 7, 16, 29, 37, 42, 54, 62, 74, 84, 85, 89, 102, and 104 are missing information such as source title, pages, article number, editorial house, place of publishing, editors, URL, or type of thesis. Also, references 18, 19, 22, 32, 34, 40, 43, 46, 48-50, 53, 54, 56, 57, 60, 64, 66, 71, 73, 75, 77, 81, 97 have source titles in sentence case instead of having each world capitalized. Reference 17 has the source title abbreviated instead of spelled out.

Specific comments

Line 28. Describe PMC.

Line 68 and 794. Change “specificity” for “selectivity”; meloxicam inhibits COX-1 too.

Lines 84-85. The pharmacokinetics of NSAIDs are species- and age-dependent; consider using references for calves. The cited references [15, flunixin in sheep; 18, dairy cows; and 19, sheep) are not appropriate.

Line 88. Reference 15 deals with flunixin in sheep, not meloxicam in cattle.

Line 116. Replace “were” for “was”.

Line 176. Remove “(HPLC)”; the abbreviation is not needed.

Line 221. Include “PCW =” at the beginning of the equation.

Line 241 and 256. For consistency (line 201), refer to “temperature” as “wound temperature”.

Lines 252 and 256. Here you mention “scrotal diameter” but in the Results, “scrotal diameter change” is provided; please be consistent and amend as appropriate.

Line 257. Please delete “and 0.05<P<0.1 was considered a statistical trend”.

Line 350. Include “PMC;” just after the opening bracket.

Lines 352-354. Delete this sentence; data are for the ML group only.

Figures 5-7. Consider tabulating these results instead of using bar graphs for each treatment. This would allow for much easier comparisons between treatments. If graphs are kept, consider using colors to differentiate scores.

Lines 414-416. Please describe SHC results by treatment too.

Lines 419-420. This sentence should be moved under subheading 3.4.3. (line 436).

Figure 8. Please indicate the units of the values in the Y axis and indicate measures of dispersion for each data point in the graph. Also, delete the sentence in line 425-426; this information is not needed.

Lines 461-463. Delete this sentence; it repeats information in lines 458-460.

Lines 468-469. Delete “normal or”.

Line 479. Provide values and references to support the “Presumed therapeutic concentrations of meloxicam”

Line 480. According to figure 1, shouldn’t be 9 days instead of 10 days?

Line 484-486. This statement may be an overinterpretation of the results as no appropriate way to measure pain was used in the study.

Line 491. Change “measurement data” for “scores”.

Line 510. If keeping this information, provide units for each of the values.

Lines 533-534. References 36 and 35 seem to be interchanged.

Line 552. Slower than what?

Line 559-560 and 816-816. Use PCM instead of the full description.

Lines 561-562, 568-569, and 744-746. These statements are not supported by your data and should be deal with as a probable, rather than a definitive, cause. Other possibility would be individual differences in the biodisposition of meloxicam via the medicated molasses lick blocks.

Line 573. Delete “(Hp)” and the use of this abbreviation thereafter. Haptoglobin has been used throughout the text until this time and there is no need to use another abbreviation.

Line 675. Provide reference in numerical order for “Olson et al’s study”.

Line 760. Provide reference in numerical order for “Van der Saag et al (2018b)”.

Lines 775-779. I guess, this would have been similar for all other treatments too. Please clarify this and/or amend as appropriate.

Lines 816. “several”? Please clarify how many and this should be reported in the Results too.

Author Response

Thank you so much for your comprehensive review of my report, the level of detail is very much appreciated!

Comment 1: Although the experimental design seems to be appropriate, it could have been improved. For instance, the need of including a surgical castration (NC) group is questionable. This is considering the vast information on behavioral and wound changes associated with castration in calves. Including such a group seems more of a potential welfare concern than a valid refinement of the study design. Having a true sham castrated group, a positive control group (SC meloxicam) and a test group (meloxicam in molasses lick blocks) would have sufficed. Furthermore, data from only 5 animals in the NC group were analyzed due to differences in castration techniques and, as discussed (e.g., line 493), this group was likely underpowered to provide any meaningful comparisons. The authors should consider whether to remove this group completely from the analysis. Also, the use of steers as a “no castration” group (line 110) seems inappropriate. Using bull calves in the PC group, and thus having a true sham castration group, would have been a much better option. This, at least, should have been discussed as a limitation of the study.

Response 1: The negative control was included as this group recieved current industry best practice pain mitigation. We felt it was important to include, especially in the contect of inflammation. I have listed the use of steers as PC as a limitation. 

Comment 2: As recognized by the authors, pain assessment in animals is challenging. Proxy measures such as animal production outcomes (e.g., liveweight) are not specifically indicative of pain. However, evaluating changes of species- and pain-specific behaviors using validated instruments is one of the most robust ways to assess pain in animals. Validated cattle pain scales specific for castration are available and the authors are aware of them (reference 81; see also Animals 13: 364, 2023; and Sci Rep 14: 15257, 2024). Constructing and using a bespoke ethogram without validating it prior to performing the study is a major drawback. What was the reasoning of using this ethogram over a validated species- and pain-specific scale? This, at least, should be discussed.

Response 2: I have listed the references from which the ethogram was adapted. 

Comment 3: In a similar way, the authors are aware of the large number of animals and long time needed to detect differences in liveweight (lines 507-509). Despite that, this parameter was measured without even blocking for this parameter wen assigning animals to the different groups. Unsurprisingly, no significant differences were found between treatment groups of 5-10 animals during the 2-week duration study. Discussing liveweight (lines 488-515) seems meaningless and deters from what I believe it should be the focus of the paper (i.e., delivery of meloxicam in the molasses lick blocks).

Response 3: I agree and have removed weight datat

Comment 4: Please clarify based on what a sample size of 10 animals per group was selected. Was an a priori power analysis performed? If so, please provide details.

Response 4: included (line 131-132)

Comment 5: The reasoning for using digital photographs to assess wound healing instead of real-time assessments needs to be clarified. Details about how the photographs were acquired and their quality, and the resolution of the screens used for displaying the photographs for assessment are required. Also, the training/experience of each of the 3 scorers with post-castration assessment in calves needs to be described. Was a formal training session provided, ideally before starting the trial, to the 3 scorers to homogenate assessments? Clarify whether wound scorers were blinded to treatments.

Response 5: updated in methods.

Comment 6:More details about the molasses lick blocks used in the study are also necessary. Please describe the weight of each block and how many blocks were available in each pen. Other than at day 6 for the ML group, were blocks replaced during the study? The way that the meloxicam was incorporated in the molasses lick blocks is also relevant as it is the quality control data for each of the blocks. Was there any testing performed to confirm that for each kg of molasses lick block 1 g of meloxicam was contained and that the NSAID was evenly distributed throughout the lick block? Also, why was the weight of the molasses lick blocks measured only for the ML group but not for the other groups? This even more considering that differences in body weight were assessed.

Response 6: Updated

Comment 7: Details about the performance in your lab for the HPLC assay to determine meloxicam and the immunoperoxidase assay to determine haptoglobin are needed. These should include limits of detection and quantification, and coefficients of variation.

Response 7: Updated expect coefficients of variation, still awaiting reply from lab that performed analysis

Comment 8: For clarity, please include and match the combined behaviors described in lines 279-287 with those detailed in Table 2. Also, please explain how is it that eating behavior was ‘amalgamated’ with other ‘normal’ behaviors (line 286) and it was then presented separately from any other behaviors (lines 442-452). Also, why data for eating on day -1 (line 232) were not presented in Figure 8 or Table 6? Consider deleting Figure 8; the same data are depicted, in a much tidier way, in Table 6. If the latter suggestion is followed, incorporated lower case superscripts to Table 6.

Response 8: Decided to remove "normal" and "abnormal" as behaviours included were influenced by factors other than pain (e.g. tail swishes for flied etc.) and was therefore unreliable. Eating updated. 

Comment 9: Please indicate the breed of the animals used in the study as well as the source of the animals. Were the calves breed on the University farm or were bought somewhere and transported to the University farm?

Response 9: Updated

Comment 10: It seems that some parameters were not measured before castration at day -1 (e.g., scrotal diameter, would temperature). When comparing the effects of treatment over time, having baseline measurements is appropriate. This would hopefully provide evidence that all groups are similar to each other before starting the study. Thus, any differences after treatment could be ruled out from groups being different to start with. This at least should be discussed as a limitation of the study.

Response 10: have listed temperature as a limitation, scrotal diameter when intact vs castrated would not be comparable. 

Comment 11: Consider including a timeline of procedures and measurements to summarize the events the animals underwent during the study. This should include the time that the first and last calf were castrated/‘sham’castrated.

Response 11: included

Comment 12: The discussion is too long and should be more focus on the main findings of the study. Some information does not add anything to the discussion of your results (e.g., lines 573-580, 604-607, 791-809) or repeats what it has been stated somewhere else (lines 610-617) and could be deleted. Although some limitations of the study are discussed, other important ones are missing and should be included. For instance, no true sham castration group, the effect of high infection rate (27%; 8 out of 30 calves) post-castration on your results, the age of the calves used and the potential different results in younger animals (J Vet Pharmacol Therap 47 (2): 143-149), the lack of technological aids (sensors) to assess activity, including grazing and ruminating activity, etc.

Response 12: have moved all limitations to end of discussion, simplified and removed weight analysis

Comment 13: There is an excessive use of references and not all of them are necessary (e.g., for the same concept two references are provided in line 68 and then two other references in line 794; for discussing haptoglobin results in lines 573-602, 21 new references were cited). Primary sources of information should be cited rather than works citing the primary sources (e.g., in lines 521 and 547, references 18 and 32 used cows and calves, not horses or dogs). The need of citing more than 1 reference for basic concepts (e.g., line 577) should be avoided.

Response 13: updated

Comment 14: Consider using a reference manager software and make sure that all cited references are formatted according to the journal specifications. For instance, references 4, 6, 7, 16, 29, 37, 42, 54, 62, 74, 84, 85, 89, 102, and 104 are missing information such as source title, pages, article number, editorial house, place of publishing, editors, URL, or type of thesis. Also, references 18, 19, 22, 32, 34, 40, 43, 46, 48-50, 53, 54, 56, 57, 60, 64, 66, 71, 73, 75, 77, 81, 97 have source titles in sentence case instead of having each world capitalized. Reference 17 has the source title abbreviated instead of spelled out.

Response 14: Updated

Reviewer 2 Report

Comments and Suggestions for Authors

Dear authors,

The document is well written. Fluent and enjoyable in all its points.

The conceptualisation of the study in my opinion is also of a high standard. Very interesting in my opinion.

I honestly have 3 main concerns that leave me questioning the reproducibility and structure of the paper:

1) To standardise the work, it is essential to assess pre-surgical pain and to verify that there are no significant differences between groups.

Secondly, how was the intraoperative nociceptive assessment performed? How can you ensure that the animals did not react to the nociceptive stimulus? You did not actually monitor the procedure. The response is extremely subjective from animal to animal and therefore also the generation of the inflammatory process. How would you justify this step?

3) Subjects treated with local anti-inflammatories because there is an underlying infection, how can you expect them to have the same type of pain as animals in which the course is normal? It would have made more sense to exclude them in my opinion.

I hope my comments are helpful in improving the quality of your document.

Thank you,

Good work

Author Response

Thank you so much for you review, your comments are greatly appreciated!

Comment 1: To standardise the work, it is essential to assess pre-surgical pain and to verify that there are no significant differences between groups.

Response 1: I have included day -1 eating behaviour, as the only treatment x time significant behaviour

Comment 2: Secondly, how was the intraoperative nociceptive assessment performed? How can you ensure that the animals did not react to the nociceptive stimulus? You did not actually monitor the procedure. The response is extremely subjective from animal to animal and therefore also the generation of the inflammatory process. How would you justify this step?

Response 2: we did not measure intra-operative pain, all calves were sedated and administered local anaesthesia to the scrotum and spermatic cords. We were more interested in pain and inflammation beyond the acute period.

Comment 3: Subjects treated with local anti-inflammatories because there is an underlying infection, how can you expect them to have the same type of pain as animals in which the course is normal? It would have made more sense to exclude them in my opinion.

Response 3: Removing the 8 animals that were treated for infection would have seriously diminished the statistical power of the experiment. This trial was designed to mimic a typical production setting to understand how this treatment would work in a commercial system.

Many thanks

Reviewer 3 Report

Comments and Suggestions for Authors

thank you for a well written and thorough project!

I have just a couple of questions. Is there a time limit to how long you would leave the meloxicam lick out and available for the calves? Is there a minimum time you would have it available for the best effect?

Is there any withdrawal time for meloxicam in cattle?

Would you recommend having the meloxicam lick available to the calves one to two days prior to the surgery? Would that improve the decrease in tissue inflammation if the meloxicam was on board sooner?

Author Response

Thank you so much for your review, your comments are greatly appreciated!

Comment 1: Is there a time limit to how long you would leave the meloxicam lick out and available for the calves? Is there a minimum time you would have it available for the best effect?

Response 1: I have now included the mean, SD, min and max PMC. The lick was avavilable for 19 hours pre-castration and the minimum PMC was 0 and the highest was 3.47. This suggests that access or aversion to the lick blocks is more of an issue than time. Future research should consider monitoring lick intake to understand these factors better. But in a practical and cost effective context, having the lick blocks out the night before castration is probably going to be the only feasible option. 

Comment 2: Is there any withdrawal time for meloxicam in cattle?

Response 2: This is a very valid question, the withholding period of metacam in cattle is 8 days, but this will definitely need further research along with toxicity. I have included this is future research.

Comment 3: Would you recommend having the meloxicam lick available to the calves one to two days prior to the surgery? Would that improve the decrease in tissue inflammation if the meloxicam was on board sooner?

Response 3: I believe this would be advantageous if practical for producers, while we didn't analyse individual results in the ML group, despite the wide variation, the small improvements seen were likely attributed to higher and continued doses of meloxicam. 

Many thanks!

Round 2

Reviewer 1 Report

Comments and Suggestions for Authors

Thank you for addressing my previous comments and incorporating many of them in your revised paper, which is much improved. I do, however, have some further comments. 

Despite removing body weight measurements (lines 272-278) weight is still mentioned in the statistical analyses section (lines 293-325) and the discussion (i.e., lines 943-945, 986-988). This should be amended and should also consider that the information on weight and PMC is relevant and it should be kept (lines 422-423, 656-658). Please consider that the information presented in the M&M must be in line with what is reported in the results and then discussed. 

I do still believe that the discussion is too long and there are too many references used throughout the paper. However, this is now more editorial than scientific.

Minor specific comments:

Lines 91-92, 1004. Replace specificity for selectivity. Meloxicam inhibits COX-2 at lower concentrations than COX-1; however, COX-1 can still be inhibited by this drug.

Line 108. Reference 18 does not support this statement; please delete it.

Line 140. Please describe the “primary outcomes to be measured” and provide details for the data used in the power calculations, including SD, and power and alpha values. Reference/s from where the data were obtained should be included.

Line 155. Please describe the type of hay offered to the animals and its nutritional composition, if available. Line 870. Please describe the type of pasture available in the paddocks and provide its nutritional composition, if available.

Line 225. The table below this line needs a title and it may be better positioned in the results rather than in the M&M section.

Lines 153, 163. Please be consistent in using the same terminology throughout the manuscript: paddocks vs. pens.

Line 405. In Table 4, there are reported values below the LOQ of 0.048 microg/ml (line 216); please amend your results as appropriate.

Lines 414-416. Delete this sentence as these definitions are not applicable in this table.

Lines 446-449. Although the two castration methods are described in the M&M, some description should be provided here too; mentioning “initial” and “latter” methods is not enough. This is to avoid any confusion between castration methods.

Line 493-494. This sentence should be moved under the following subheading (i.e., 3.3.3).

Line 497. Please indicate the units in the y axis of Figure 5.

Lines 567. Consider changing “indicates” for “suggests”.

Line 605. PMC has been previously defined; please amend as appropriate.

Lines 618. References 38 and 39 are inverted; please check and amend as appropriate.

Line 676. Instead of using another abbreviation (i.e., Hp), consider spelling haptoglobin in full throughout the manuscript.

Lines 676-677. For consistency, use the abbreviations that describe these two groups of calves.

Line 679. SHC has been previously defined; please amend as appropriate.

Lines 685-692. Consider moving this section at the beginning of the paragraph (i.e., line 676).

Line 905. Provide a continuous numerical reference for Van der Saag et al (2018b).

Line 1002. Although with a 10-day course administration of a non-selective COX inhibitor, abomasal ulcers and renal failure have been reported in cattle (J Vet Pharmacol Ther 39 (5): 518-521, 2016).

Although more editorial than scientific, references should be cited as per the journals requirements. For instance, the source for some references is missing (e.g., line 1099), sources are provided in different formats (e.g., Sentence case vs. Each Word Capitalized).

Author Response

Thank you for your prompt revision and recommendations!

Comment 1: Despite removing body weight measurements (lines 272-278) weight is still mentioned in the statistical analyses section (lines 293-325) and the discussion (i.e., lines 943-945, 986-988). This should be amended and should also consider that the information on weight and PMC is relevant and it should be kept (lines 422-423, 656-658). Please consider that the information presented in the M&M must be in line with what is reported in the results and then discussed. 

Response 1: I have added detail about weights recorded on day -1 in the treatments and experimental design section

Comment 2: I do still believe that the discussion is too long and there are too many references used throughout the paper. However, this is now more editorial than scientific.

Response 2: We have discussed this as a research group and believe that the length of the discussion is equivalent to other papers and feel it necessary to highlight the issues raised in the research.

Minor specific comments have been amended.

Many thanks!